# The charged $Z_c$ and $Z_b$ structures in a constituent quark model approach

**Pablo García Ortega[1]\***, **Jorge Segovia[2]**,
**David Rodríguez Entem[1]** and **Francisco Fernández[1]**

**1** Grupo de Física Nuclear and Instituto Universitario de Física Fundamental y Matemáticas
(IUFFyM), Universidad de Salamanca, E-37008 Salamanca, Spain
**2** Departamento de Sistemas Físicos, Químicos y Naturales,
Universidad Pablo de Olavide, E-41013 Sevilla, Spain

\* pgortega@usal.es

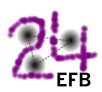 *Proceedings for the 24th edition of European Few Body Conference,*
## Abstract

**The nature of the recently discovered $Z_c$ and $Z_b$ structures is intriguing. Their charge forces its minimal quark content to be $Q\bar{Q}q\bar{q}$ (where $Q = \{c, b\}$ and $q = \{u, d\}$). In this work we perform a molecular coupled-channels calculation of the $I^G(J^{PC}) = 1^+(1^{+-})$ charm and bottom sectors in the framework of a constituent quark model which satisfactorily describes a wide range of properties of (non-)conventional hadrons containing heavy quarks. All the relevant channels are included for each sector, i.e.: The $D^{(*)}\bar{D}^*+h.c.$, $\pi J/\psi$ and $\rho\eta_c$ channels for the $Z_c$ and $B^{(*)}B^*$ and $\Upsilon(nS)\pi$ ($n = 1, 2, 3$) channels for the $Z_b$ analysis. Possible structures of these resonances will be discussed.**



## 1 Introduction

The search of exotic structures in the heavy meson spectra, beyond the simplest $q\bar{q}$ structure, is relatively recent. In 2003, the first and most iconic resonance, the so-called $X(3872)$, was spotted by Belle [1] and promptly confirmed by other B-factories and accelerator facilities such as BaBar, CDF and D0 Collaborations. At the same time, BaBar and CLEO Collaborations reported the discovery of the $D_{s0}^*(2317)$ and $D_{s1}(2460)$ [2, 3], two puzzling structures in the heavy-light mesons spectrum. Even though the properties of such resonances where compatible with a $q\bar{q}$ quark content, they were difficult to accommodate in the naive quark model due to their unexpected masses and decay properties, which pointed to a non-negligible contribution of higher Fock-state components.

The clearest evidence of exotic structures appeared in 2011, when the Belle Collaboration [4, 5] announced the observation of two meson-like structures in the bottom sector

with forbidden quantum numbers for a $b\bar{b}$ pair. The so-called $Z_b(10610)$ and $Z_b(10650)$ where charged structures close to the $B\bar{B}^*$ and $B^*\bar{B}^*$ thresholds, respectively, spotted in the $\Upsilon(5S) \to \pi^+\pi^-\Upsilon(nS)$ reaction. Few years later, their charmonium partners arrived. The BESIII and Belle Collaborations discovered the $Z_c(3900)$ [6,7], close to the $D\bar{D}^*$ threshold, in the $\pi^+\pi^-J/\psi$ invariant mass spectrum of the $e^+e^- \to \pi^+\pi^-J/\psi$ process at $\sqrt{s} = 4.26$ GeV. The spin and parity of the charged $Z_c(3900)$ was set to $J^P = 1^+$ [8], with a mass $M = (3881.2\pm4.2\pm52.7)$ and width $\Gamma = (51.8\pm4.6\pm36.0)$ MeV. Additionally, the $Z_c(4020)$ resonance, close to the $D^*\bar{D}^*$ threshold, was discovered soon after at BESIII in the $e^+e^- \to \pi^+\pi^-h_c$ channel with a mass of $M = (4022.9 \pm 0.8 \pm 2.7)$ MeV/$c^2$ and a width of $\Gamma = (7.9 \pm 2.7 \pm 2.6)$ MeV [9].

All the previous $Z_b$ and $Z_c$ states are charged, so they cannot be described as pure $q\bar{q}$ states. Their closeness to $B^{(*)}\bar{B}^*$ and $D^{(*)}\bar{D}^*$ thresholds indicate a dominant meson-meson component in their wave functions. These features have been widely explored in different theoretical scenarios, such as hadron molecules [10–13], tetraquark structures [14–17] or simple kinematic effects linked to the opening of meson-meson thresholds [18,19].

In this work we explore the $I^G(J^{PC}) = 1^+(1^{+-})$ charm and bottom sector in a coupled-channels scheme, including the closest meson-meson thresholds. The meson-meson interaction is described in the framework of a constituent quark model [1] successfully employed to explain the meson and baryon phenomenology from the light to the heavy quark sector (see for example Ref. [20, 22–24]. Moreover, the $D^{(*)}\bar{D}^*$ residual interaction deduced from the model has been satisfactorily used to describe meson-meson [25–27] molecular states.

## 2 Theoretical Formalism

Invariance under chiral rotations is a symmetry of the Quantum Chromodynamics (QCD) Lagrangian with massless light quarks. However, this symmetry is not satisfied in nature, pointing to a spontaneous breaking within QCD. This effect has several interesting consequences, one of those being the emergence of a momentum-dependent constituent quark mass, $M = M(q^2)$ and $M(q^2 \to 0) = m_q$, with $m_q$ the current quark mass, and the appearance of Goldstone bosons which mediate the interaction among light quarks.

The latter phenomenology can be synthesized in a constituent quark model (CQM), which is described by the following low-energy Lagrangian [28]

$$\mathcal{L} = \bar{\psi}(i\,\partial\!\!\!/ - M(q^2)U^{\gamma_5})\psi, \tag{1}$$

where $U^{\gamma_5} = e^{i\lambda_a\phi^a\gamma_5/f_\pi}$ is the Goldstone-boson field matrix. This matrix can be expanded as

$$U^{\gamma_5} = 1 + \frac{i}{f_\pi}\gamma^5\lambda^a\pi^a - \frac{1}{2f_\pi^2}\pi^a\pi^a + \dots, \tag{2}$$

so the first term can be identified as the constituent quark mass, the second one describes the pseudoscalar meson exchange interaction among quarks and the third term, whose main contribution is the two-pion exchange, can be coded by means of a scalar-meson exchange potential.

The model is completed with QCD non-perturbative effects such as the confinement interaction, implemented phenomenologically so colored hadrons are prohibited. Within our CQM, the confinement is modelled with a linear-rising potential, due to multi-gluon exchanges among quarks, which is screened at large inter-quark distances due to sea quarks [29]:

$$V_{\mathrm{CON}}(\vec{r}) = \left[-a_c(1 - e^{-\mu_c r}) + \Delta\right](\vec{\lambda}_q^c \cdot \vec{\lambda}_{\bar{q}}^c). \tag{3}$$

---

[1]The interested reader is referred to Refs. [20, 21] for detailed reviews about the quark model in which this work is based.

Here, $a_c$ and $\mu_c$ are model parameters. From the latter equation it is easy to see that the potential is linear at short inter-quark distances with an effective confinement strength $\sigma = -a_c \mu_c (\vec{\lambda}_i^c \cdot \vec{\lambda}_j^c)$, becoming constant at large distances.

Beyond the non-perturbative energy scale, the dynamics of the quarkonium is expected to be dominated by QCD perturbative effects. That is taken into account by means of the one-gluon exchange potential, derived from the following Lagrangian

$$\mathcal{L}_{qqg} = i\sqrt{4\pi\alpha_s}\,\bar{\psi}\gamma_\mu G_a^\mu \lambda^a \psi. \tag{4}$$

Here, $\alpha_s$ is the strong coupling constant, $\lambda^a$ are the $SU(3)$ colour matrices and $G_a^\mu$ is the gluon field. To consistently treat the light- and heavy-quark sectors we employ a gluon coupling constant that scales with the reduced mass of the interacting quarks.

Explicit expressions, model parameters and a detailed physical background of the constituent quark model used in this work can be found in, e.g., Refs. [20, 23]

These CQM describes the interaction among constituent quarks. When dealing with meson (hadron) degrees of freedom, the Resonating Group Method [30] can be employed to obtain the interaction at meson level from the microscopic interaction at quark level. In RGM, mesons are considered as quark-antiquark clusters and an effective cluster-cluster interaction emerges from the underlying $q\bar{q}$ dynamics.

The meson eigenstates $\phi_C(\vec{p}_C)$ of a general meson $C$, with $\vec{p}_C$ the relative momentum between the quark and antiquark of the meson $C$, are calculated by means of the two-body Schrödinger equation using the Gaussian Expansion Method [31]. We use Gaussian trial functions with ranges given by a geometrical progression [31], which optimizes the method and reduces the number of free parameters. This choice produces a dense distribution at short distances, enabling a better description of the dynamics mediated by short range potentials.

The orbital wave function of a system composed of two mesons $A$ and $B$ with distinguishable quarks can be, then, written as[2]

$$\langle \vec{p}_A \vec{p}_B \vec{P} \vec{P}_{\text{c.m.}} | \psi \rangle = \phi_A(\vec{p}_A)\phi_B(\vec{p}_B)\chi_\alpha(\vec{P}), \tag{5}$$

where $\chi_\alpha(\vec{P})$ is the relative wave function between the two clusters, and $\alpha$ labels the set of quantum numbers needed to uniquely define a certain partial wave.

Such relative wave function can be obtained as the solution of the projected Schrödinger equation:

$$\left(\frac{\vec{P}'^2}{2\mu} - E\right)\chi_\alpha(\vec{P}') + \sum_{\alpha'}\int \left[{}^{\text{RGM}}V_D^{\alpha\alpha'}(\vec{P}',\vec{P}) + {}^{\text{RGM}}V_R^{\alpha\alpha'}(\vec{P}',\vec{P})\right]\chi_{\alpha'}(\vec{P})\,d\vec{P} = 0, \tag{6}$$

where $E$ is the energy of the system. As we can see, two types of diagrams appear: one which does not consider quark exchanges between clusters, called *direct* potential; and one involving quark exchanges, dubbed *rearrangement* potential.

The direct potential ${}^{\text{RGM}}V_D^{\alpha\alpha'}(\vec{P}',\vec{P})$ of a reaction $AB \to CD$ can be written as

$$ {}^{\text{RGM}}V_D^{\alpha\alpha'}(\vec{P}',\vec{P}) = \sum_{i,j}\int d\vec{p}_A\,d\vec{p}_B\,d\vec{p}_C\,d\vec{p}_D\,\phi_C^*(\vec{p}_C)\phi_D^*(\vec{p}_D)V_{ij}^{\alpha\alpha'}(\vec{P}',\vec{P})\phi_A(\vec{p}_A)\phi_B(\vec{p}_B), \tag{7}$$

where $\{i,j\}$ runs over the constituents of the involved mesons. The quark rearrangement potential ${}^{\text{RGM}}V_R^{\alpha\alpha'}(\vec{P}',\vec{P})$ represents a natural way to connect meson-meson channels with dif-

---

[2]For simplicity, we have dropped off the spin-isospin wave function, the product of the two color singlets and the wave function that describes the center-of-mass motion.

ferent quark content, such as $\pi J/\psi$ and $D\bar{D}^*$. It can be calculated with

$$
{}^{\text{RGM}}V_R^{\alpha\alpha'}(\vec{P}',\vec{P}) = \sum_{i,j} \int d\vec{p}_A\, d\vec{p}_B\, d\vec{p}_C\, d\vec{p}_D\, d\vec{P}''\, \phi_A^*(\vec{p}_C) \times
$$

$$
\times\, \phi_D^*(\vec{p}_D)V_{ij}^{\alpha\alpha'}(\vec{P}',\vec{P}'')P_{mn}\left[\phi_A(\vec{p}_A)\phi_B(\vec{p}_B)\delta^{(3)}(\vec{P}-\vec{P}'')\right], \tag{8}
$$

where $P_{mn}$ is the operator that exchanges quarks between clusters.

The solution of the RGM coupled-channels equation is performed deriving from Eq. (6) a set of coupled Lippmann-Schwinger equations of the form

$$
T_\alpha^{\alpha'}(E;p',p) = V_\alpha^{\alpha'}(p',p) + \sum_{\alpha''}\int dp''\,p''^2\, V_{\alpha''}^{\alpha'}(p',p'')\frac{1}{E - \mathcal{E}_{\alpha''}(p'')}\, T_\alpha^{\alpha''}(E;p'',p). \tag{9}
$$

Here, $V_\alpha^{\alpha'}(p',p)$ is the projected potential that contains the direct and rearrangement potentials, and $\mathcal{E}_{\alpha''}(p'')$ is the energy corresponding to a momentum $p''$, written in the non-relativistic case as:

$$
\mathcal{E}_\alpha(p) = \frac{p^2}{2\mu_\alpha} + \Delta M_\alpha, \tag{10}
$$

where $\mu_\alpha$ is the $AB$-system reduced mass corresponding to the channel $\alpha$, and $\Delta M_\alpha$ is the difference between the threshold of the $AB$ system and the reference one.

The coupled-channels Lippmann-Schwinger equation is solved via the matrix-inversion method proposed in Ref. [32], generalized in order to include channels with different thresholds. From the full $T$-matrix, it is direct to extract the on-shell part, directly related to the scattering matrix as

$$
S_\alpha^{\alpha'} = 1 - 2\pi i\sqrt{\mu_\alpha\mu_{\alpha'}k_\alpha k_{\alpha'}}\, T_\alpha^{\alpha'}(E+i0^+;k_{\alpha'},k_\alpha), \tag{11}
$$

in non-relativistic kinematics, with $k_\alpha$ the on-shell momentum for channel $\alpha$.

The final goal of the study is the analysis of the existence of states above and below thresholds within the same formalism. For that purpose, all the potentials and kernels are analytically continued for complex momenta so the poles of the $T$-matrix in any possible Riemann sheet can be detected.

## 3 Results

### 3.1 $Z_c(3900)$ and $Z_c(4020)$ structures

The aim of the present study is to use the constituent quark model of Ref. [20] as a basis to perform a coupled-channels calculation of the $I^G(J^{PC}) = 1^+(1^{+-})$ sector with hidden charm and bottom. For the charmonium sector, we include the closest thresholds to the $Z_c(3900)$ and $Z_c(4020)$ experimental masses: $\pi J/\psi$ (3234.19 MeV/$c^2$), $\rho\eta_c$ (3755.79 MeV/$c^2$), $D\bar{D}^*$ (3875.85 MeV/$c^2$), $D^*\bar{D}^*$ (4017.24 MeV/$c^2$), where the threshold masses are shown in parenthesis. The $h_c\pi$ channel is not considered as it couples weakly to the $D^{(*)}\bar{D}^*$ channels. In fact, the only contribution of this channel would come from the $^3D_1$ component of the internal wave function of the $D^*$ meson, which in our model is $\sim 0.03\%$ and it is neglected in the present calculation. For the $^3S_1$ component of the $D^*$, the coupling to $h_c\pi$ is exactly zero. Similarly, we do not include other nearby channels such as the $\chi_{cJ}\rho$ ones, whose couplings are found to be almost three orders of magnitude smaller than those of the $J/\psi\pi$ and $\eta_c\rho$ channels.

The invariant mass distribution of the $D\bar{D}^*$, $\pi J/\psi$ and $D^*\bar{D}^*$ channels in the reaction $e^+e^- \to \pi^\pm(AB)^\mp$ [3] predicted by our model are shown in Figs. 1 and 2, following the procedure

---

[3]Where $AB$ is the final channel under consideration.

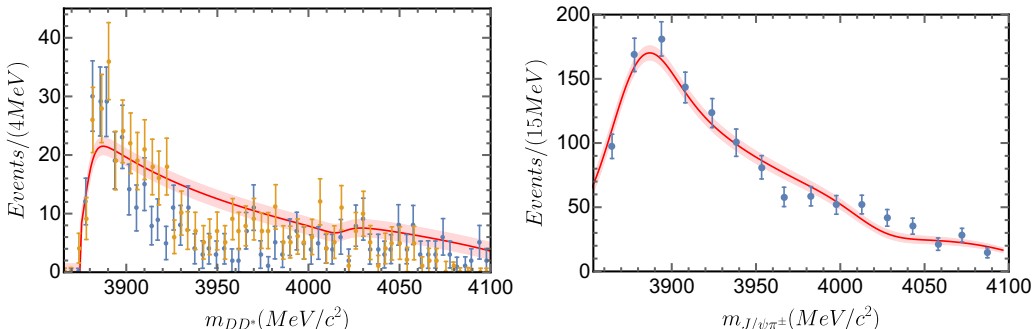

Figure 1: Line shapes for $D\bar{D}^*$ (left panel) and $\pi J/\psi$ (right panel) invariant mass spectrum at $\sqrt{s} = 4.26$ GeV for the reactions $e^+e^- \to \pi^{\pm}(D\bar{D}^*)^{\mp}$ and $e^+e^- \to \pi^+\pi^-J/\psi$, respectively. Experimental data are from Ref. [8,34], respectively. The theoretical line shapes have been convoluted with the experimental resolution. The line-shape's 68% uncertainty is shown as a shadowed area.

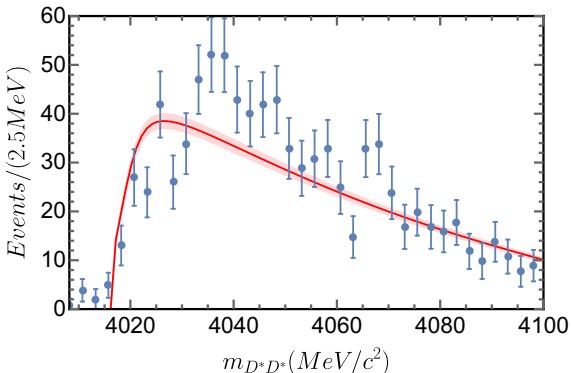

Figure 2: Line shape for $D^*\bar{D}^*$ at $\sqrt{s} = 4.26$ GeV. Experimental data are from Refs. [35]. The theoretical line shapes have been convoluted with the experimental resolution. The line-shape's 68%-uncertainty is shown as a shadowed area.

of Ref. [33]. Normalization factors $\mathcal{N}_{AB}$ and production amplitudes $\mathcal{A}_{AB}$, that appear in the line shapes, are obtained from a fit performed with the experimental results of Refs. [6,8,34], that is, experimental data for $D\bar{D}^*$ and $\pi J/\psi$ channels. The amplitudes $\mathcal{A}_{AB}$ obtained from such fit are, then, employed for the $D^*\bar{D}^*$ channel in order to obtain the global normalization $\mathcal{N}_{D^*\bar{D}^*}$ from experimental data of Ref [35]. In order to describe the experimental measurement, the theoretical line shapes have been convoluted with the detector resolution.

In order to optimize the fit, only $D\bar{D}^*$ experimental data up to 3.92 GeV was considered. For larger energies the background is expected to dominate [36]. The $\pi J/\psi$ data of Ref. [8] was fitted from 3.85 GeV on. In the theoretical line shape of the $D\bar{D}^*$ channel one can clearly see two enhancements related with the opening of the $D\bar{D}^*$ and $D^*\bar{D}^*$ thresholds, which can be associated with the $Z_c(3900)$ and the $Z_c(4020)$. In the $\pi J/\psi$ case, only one enhancement appears around 3.87 GeV while the opening of the $D^*\bar{D}^*$ channel appears in our theoretical line shape as a slight step down in the number of events.

Apart from the unknown $\pi + A + B$ vertex details, encoded in the fitted normalization and amplitude factors, our CQM is able to reproduce the experimental data with no fine-tuning of the parameters. Attending to the nature of the $Z_c(3900)$ and the $Z_c(4020)$ states, we have examined the pole structure of the $S$-matrix for different coupled-channels calculations. Our results are shown in Table 1. For the $Z_c(3900)$, even for a one-channel $D\bar{D}^*$ calculation, the

Table 1: The $S$-matrix pole positions, in MeV/$c^2$, for different coupled-channels calculations. The included channels for each case are shown in the first column. Poles are given in the second and fourth columns by the value of the complex energy in a specific Riemann sheet (RS). The RS columns indicate whether the pole has been found in the first (F) or second (S) Riemann sheet of a given channel. Each channel in the coupled-channels calculation is represented as an array's element, ordered with increasing energy.

| Calculation | $Z_c(3900)$ pole | RS | $Z_c(4020)$ pole | RS |
|---|---|---|---|---|
| $D\bar{D}^*$ | $3871.37 - 2.17\,i$ | (S) | - | - |
| $D\bar{D}^* + D^*\bar{D}^*$ | $3872.27 - 1.85\,i$ | (S,F) | $4014.16 - 0.10\,i$ | (S,S) |
| $\rho\eta_c + D\bar{D}^*$ | $3871.32 - 0.00\,i$ | (S,S) | - | - |
| $\rho\eta_c + D\bar{D}^* + D^*\bar{D}^*$ | $3872.07 - 0.00\,i$ | (S,S,F) | $4013.10 - 0.00\,i$ | (S,S,S) |
| $\pi J/\psi + \rho\eta_c + D\bar{D}^* + D^*\bar{D}^*$ | $3871.74 - 0.00\,i$ | (S,S,S,F) | $4013.21 - 0.00\,i$ | (S,S,S,S) |

Table 2: The $Z_b$ states parameters from the $S$-matrix pole positions.

| | $Z_b(10610)^{\pm}$ | $Z_b(10650)^{\pm}$ |
|---|---|---|
| Mass | 10600.45 | 10644.74 |
| Width | 2.80 | 8.88 |
| $\mathcal{P}_{\max}$ | 95.76% ($BB^*$) | 64.85% ($B^*B^*$) |
| $\Gamma_{BB^*}$ | — | 8.68 |
| $\Gamma_{\Upsilon(1S)\pi}$ | 0.94 | 0.08 |
| $\Gamma_{\Upsilon(2S)\pi}$ | 0.65 | 0.002 |
| $\Gamma_{\Upsilon(3S)\pi}$ | 1.21 | 0.12 |

$S$-matrix shows a virtual pole below threshold, which is maintained when further channels are added. In the complete calculation, the $Z_c(3900)$ is associated with a pole located in the imaginary axis of the second Riemann sheet below the $D\bar{D}^*$ threshold and, thus, it is a virtual state. The situation is similar in the case of the $Z_c(4020)$, which is interpreted as a virtual state located below the $D^*\bar{D}^*$ threshold. Further details of the calculation can be found in Ref. [33].

## 3.2  $Z_b(10610)$ and $Z_b(10650)$ structures

For the bottomonium sector we follow the same approach as above, i.e., we perform a coupled-channels calculation of the $I^G(J^{PC} = 1^+(1^{+-})$ sector including the following channels: $B\bar{B}^*$ (10604.1 MeV/$c^2$), $B^*\bar{B}^*$ (10649.3 MeV/$c^2$), $\Upsilon(1S)\pi$ (9597.6 MeV/$c^2$), $\Upsilon(2S)\pi$ (10160.5 MeV/$c^2$) and $\Upsilon(3S)\pi$ (10492.5 MeV/$c^2$). Contrary to the $Z_c$ case, the $B^{(*)}\bar{B}^*$ interaction is strong enough to produce a real bound state below the $B\bar{B}^*$ threshold and a resonance close to the $B^*\bar{B}^*$ threshold, as can be seen in Table 2. Due to Heavy Flavour Symmetry, the $B^{(*)}\bar{B}^*$ and $D^{(*)}\bar{D}^*$ interactions are practically identical. However, the kinetic energy of the involved channels is reduced in the bottom sector due to the larger mass of the $b$-quark mass, favouring the creation of bound states.

In order to see if such predicted poles describe the experimental situation, analogously as for the $Z_c$'s states, we calculated the predicted line shapes in each channel between 10.55 and 10.75 GeV. We consider they are coming from a $\pi + A + B$ vertex with a center of mass energy of $\sqrt{s} = 10.865$ GeV, according to Refs. [4, 5, 37]. The procedure to obtain the line shapes is the same as for the $Z_c$'s (see Ref. [33]). The results can be seen in Figs. 3 for the $B\bar{B}^*$, $B^*\bar{B}^*$,

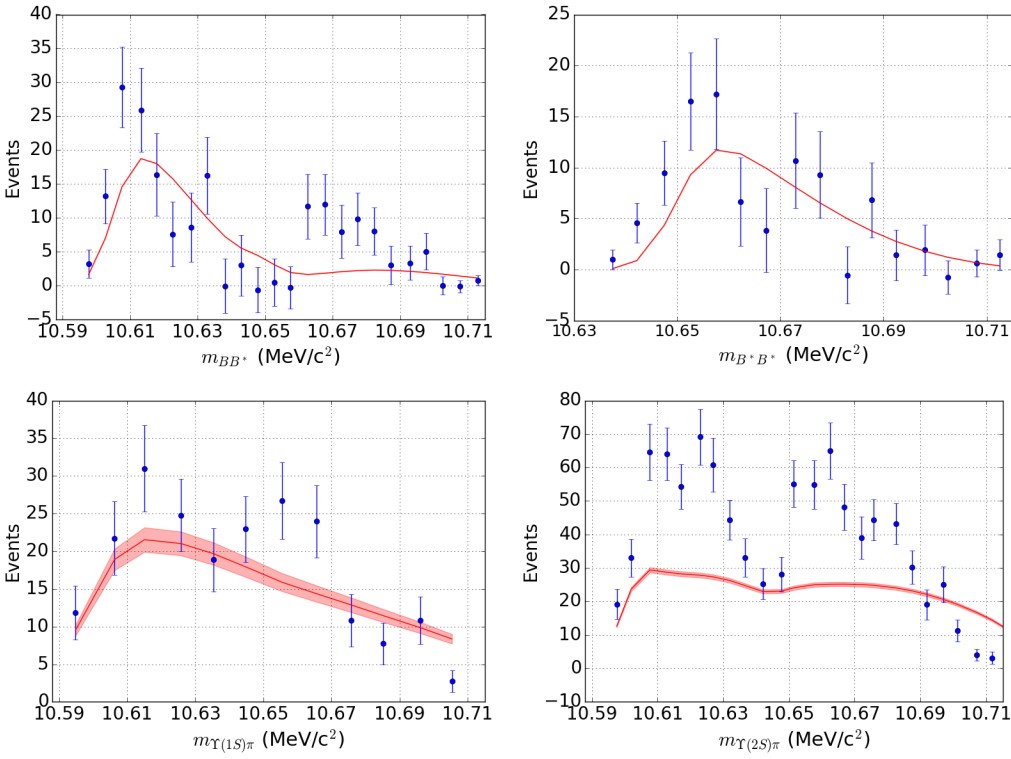

Figure 3: Line shapes for different channels. Experimental data from Refs. [4,5,37].

$\Upsilon(1S)\pi$ and $\Upsilon(2S)\pi$. The description of the $Z_b(10610)$ peak is in nice agreement with the experimental data for the $B\bar{B}^*$ and $\Upsilon(1S)\pi$ channels. The $Z_b(10650)$ peak is also properly described for the $B^*\bar{B}^*$ channel, but somehow it is missing in the $B\bar{B}^*$ and $\Upsilon(1S)\pi$ line shapes. Such discrepancy could be caused by the simple ansatz considered for the $\pi + A + B$ vertex or perhaps it points to a small non-diagonal coupling of the $B^*\bar{B}^*$ channel with the rest of the thresholds within our model which should be improved. Both peaks emerge in the $\Upsilon(2S)\pi$ line shape, but their strength is not high enough to describe the experimental situation. Further research is ongoing in order to clarify the inner structure of the $Z_b(10610)$ and $Z_b(10650)$.

## 4 Conclusion

The $Z_b$'s and $Z_c$'s are very peculiar structures, different from other molecular states of the bottomonium and charmonium spectrum. In order to clarify their inner structure, we have performed, within the framework of a constituent quark model, a coupled-channels calculation of the $I^G(J^{PC}) = 1^+(1^{+-})$ sector around the energies of the $Z_c(3900)$ and $Z_c(4020)$, for the charmonium sector, and the $Z_b(10610)$ and $Z_b(10650)$, for the bottomonium case, including the most relevant thresholds.

The line shapes of the $D^{(*)}\bar{D}^*$, $\pi J/\psi$, $B^{(*)}\bar{B}^*$ invariant mass distributions are well reproduced without any fine-tuning of the model parameters. Some channels such as the $\Upsilon(nS)\pi$ are not well understood yet and will require further investigation. The analysis of the $S$-matrix poles allows us to conclude the following. For the $Z_c$'s, the structure of the line shapes is due to the presence of two virtual states that can be seen as $D^{(*)}\bar{D}^*$ threshold bumps, and have an overall good agreement with the location of the $Z_c(3900)$ and $Z_c(4020)$ signals. These results confirm the conclusion of the lattice QCD calculation of Ref. [38]. For the $Z_b$'s, a real bound state, below the $B\bar{B}^*$ threshold, and a resonance, below the $B^*\bar{B}^*$ threshold, emerge

from the coupled-channels interactions, matching almost all the properties of the $Z_b(10610)$ and $Z_b(10650)$ structures, respectively.

# Acknowledgements

**Funding information**  This work has been partially funded by Ministerio de Economía, Industria y Competitividad under Contracts No. FPA2016-77177-C2-2-P, FPA2017-86380-P and by EU STRONG-2020 project under the program H2020-INFRAIA-2018-1, grant agreement no. 824093.

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
