# Peer review of "The charged $Z_c$ and $Z_b$ structures in a constituent quark model approach"

_SciPost Physics Proceedings, doi:SciPost Phys. Proc. 3, 013 (2020)_

## Round 1 · Referee Report · Anonymous (Referee 1) · 2019-10-28

Strengths

  1. Timely
  2. Well written
  3. Good results

Weaknesses

  1. Some confusing statements that can be amended

Report

Report SciPost Physics Proceedings The charged Zc and Zb structures in a constituent quark model approach by: P.G. Ortega, J. Segovia, D. Entem, F. Fernandez)

This work analyzes the nature of some charged structures in the chamonium and bottomonium sectors within a constituent quark model. The paper is well presented, sufficient details of the theoretical model are given and the results are clearly presented and discussed. I have only a few minor comments or typos that, when addressed properly, will leave the manuscript ready for publication: 1. Please, comment in the text (second paragraph in page 5) and in the caption of Fig. 1 the reactions to which the invariant mass distributions of the figure correspond.
2. I find strange that the widths extracted from the poles of the Zb’s are relatively narrow (2.8 and 8.9 MeV, according to Table 2), while these states seem to leave a much wider signature (20 MeV, 30 MeV) in the invariant mass distributions shown in Figure 3. Please comment. 3. Page 8, line 4: what does “point-wise” behaviour mean? 4. The authors mention in page 5, paragraph 3: “while the opening of the DDbar threshold appears as a slight step down in the number of events”. The errors prevent one to see a step down in the data... Do you mean that the model shows a step down? Please clarify.
Some typos: After Eq. 81): The Goldstone-boson field matrix 9 lines after eq. (4): and an effective cluster-cluster interaction emerges 1st line in page 4: two types of diagrams appear Page 5 paragraph 3, line 6: only one enhancement appears

Requested changes

See report above

  • validity: high
  • significance: high
  • originality: high
  • clarity: high
  • formatting: excellent
  • grammar: good

Author:  Pablo G. Ortega  on 2019-11-14  [id 647]

(in reply to Report 1 on 2019-10-28)
Category:
answer to question
correction

We thank the referee for her/his report on the submitted version of our manuscript. We have considered the referee's report and respond to the comments below. For clarity, we follow in our replies the same order as the referee's comments.

  1. We have changed the first line of the caption of Fig. 1 to

Line shapes for $D\bar D^*$ (left panel) and $\pi J/\psi$ (right panel) invariant mass spectrum at $\sqrt{s}=4.26$ GeV for the reactions $e^+e^-\to\pi^\pm(D\bar D^*)^\mp$ and $e^+e^-\to\pi^+\pi^-J/\psi$, respectively.

and the text in the second paragraph in page 5:

The invariant mass distribution of the $D\bar D^*$, $\pi J/\psi$ and $D^*\bar D^*$ channels in the reaction $e^+e^-\to \pi^\pm(AB)^\mp$ [footnote: Where $AB$ is the final channel under consideration.] predicted by our model are shown in Figs.1 and 2.

  1. Our total decay width of the Zb structure is not estimated from the signal found in the invariant mass of the relevant channels: $BB*$ for Zb(10610) and $B*B*$ for Zb(10650). This cannot be done because theoretical and experimental agreement still need to be improved. Therefore, we are reporting in Table 2 the sum of the exclusive channels. Since all the exclusive decay channels of the Zb's are not taken into account, our theoretical value of the total widths are smaller than the experimental ones. Besides, such small width is not reflected in the theoretical line shapes because the latter are convoluted with the experimental response function.

  2. For clarity, we have replaced the sentence to the following:

For the $Z_c$'s, the structure of the line shapes is due to the presence of two virtual states that can be seen as $D^{(*)}\bar D^*$ threshold bumps, and have an overall good agreement with the location of the $Z_c(3900)$ and $Z_c(4020)$ signals.

  1. We referred to the step down that appears in our theoretical line shape. We have included that information in the sentence:

the opening of the $D^*\bar D^*$ channel appears in our theoretical line shape as a slight step down in the number of events.

Finally, we have corrected the typos mentioned by the referee in his/her report.

All the changes mentioned above will be included in the new version of the manuscript.

---

## Editorial Decision

published